# Saturday Night Fever: Interpersonal Violence as a Reason for Presentation in a University Emergency Department

**DOI:** 10.3390/ijerph20054552

**Published:** 2023-03-03

**Authors:** Jason-Alexander Hörauf, Jan-Niklas Franz, Julian Zabel, Frederik Hartmann, Philipp Störmann, Ingo Marzi, Maren Janko, René D. Verboket

**Affiliations:** Department of Trauma, Hand and Reconstructive Surgery, Hospital of the Johann Wolfgang Goethe—University Frankfurt am Main, 60590 Frankfurt, Germany

**Keywords:** interpersonal violence, emergency department, alcohol

## Abstract

(1) Presentations to a trauma emergency department following a violent confrontation account for a relevant proportion of the overall population. To date, violence (against women) in the domestic setting has been studied in particular. However, representative demographic and preclinical/clinical data outside of this specific subgroup on interpersonal violence are limited; (2) Patient admission records were searched for the occurrence of violent acts between 1 January and 31 December 2019. A total of 290 patients out of over 9000 patients were retrospectively included in the “violence group” (VG). A “typical” traumatologic cohort (presentation due to, among other things, sport-related trauma, falls, or traffic accidents) who had presented during the same period served as comparison group. Then, differences in the type of presentation (pedestrian, ambulance, or trauma room), time of presentation (day of week, time of day), diagnostic (imaging) and therapeutic (wound care, surgery, inpatient admission) measures performed, and discharge diagnosis were examined; (3) A large proportion of the VG were male, and half of the patients were under the influence of alcohol. Significantly more patients in the VG presented via the ambulance service or trauma room and during the weekend and the night. Computed tomography was performed significantly more often in the VG. Surgical wound care in the VG was required significantly more often, with injuries to the head being the most common; (4) The VG represents a relevant cost factor for the healthcare system. Because of the frequent head injuries with concomitant alcohol intoxication, all mental status abnormalities should be attributed to brain injury rather than alcohol intoxication until proven otherwise, to ensure the best possible clinical outcome.

## 1. Introduction

Alongside the typical reasons for presentations, such as distortion trauma, falls or traffic accidents, presentations/admissions following violent confrontation are also part of the trauma daily routine [1]. According to the World Health Organization (WHO), violence is defined as “the actual or threatened intentional use of physical or psychological force or power directed against oneself or another person, group, or community, resulting in actual or high probability of injury, death, psychological harm, maldevelopment, or deprivation” [2]. Just as psychological treatment and care is important in the further course, primary medical care for immediate health consequences is warranted first and foremost. The presentation of this patient collective, then, often occurs via the emergency department (ED). Previous studies of interpersonal violence have focused particularly on surveying physical violence in the domestic setting, with most work examining violence against women [3]. Apart from this, insufficient representative data on interpersonal violence exist. Therefore, the aim of the present study is to investigate the interpersonal violence incidents at an urban university hospital that led to a presentation/admission to the ED. In direct comparison to “typical” reasons (distortion trauma, falls, etc.) for presentation in a trauma ED, the demographic, diagnostic, therapeutic and weekly or daily differences between the groups will be examined.

## 2. Materials and Methods

The present work is a retrospective study. Patients were selected from the hospital′s internal information system Orbis—Dedalus (Saarbrücken, Germany) who presented either on their own or were admitted via the ambulance service to the ED of the University Hospital Frankfurt am Main between 1 January and 31 December 2019. For each patient, the attending physician routinely creates an admission record. The admission records were searched for the occurrence of violent acts during the study period mentioned above. If the admission record documented that the presentation was due to a violent confrontation, the respective patient was included in the violence group. More specifically, patients were included in the violence group if they sustained visible injuries (e.g., wounds, bruises, hematomas) or demonstrable injuries (e.g., tenderness of the chest) on clinical examination in the course of a reported (self-reported or third-party) physical confrontation. Thus, a total of 290 patients were included in the violence group. Patients who presented to the ED in the context of intentional self-harm were excluded from the study. Since there are already several studies in the literature that explicitly investigated cases of women who had experienced domestic violence, this group was also excluded [4].

Subsequently, a comparison group was established. If a violent confrontation was recorded on a particular day, all other admission records generated for that day were also captured and analyzed for the same characteristics (age, gender, reason for presentation, admission via ambulance service/trauma room, radiologic imaging performed, necessity for wound care or surgical therapy, inpatient admission, discharge diagnosis and body region primarily injured by the trauma). These then served as a data pool for the comparison group. The variables were then tested graphically for normal distribution using a Q-Q plot. To increase the robustness of the results, several random samples were then drawn from this data set and the results subsequently were validated with the other parts. From the data pool, a random sample of 355 patients was defined as the comparison group using SPSS.

This study was approved by the local ethics committee (19-491) of Johann-Wolfgang-Goethe University.

### 2.1. Statistical Analysis

The collected data were first assembled in Microsoft Excel version 16.63.1 (Redmond, WA, USA) and then imported into the software program Statistical Package for Social Sciences (SPSS) version 26.0 (SPSS Inc., Chicago, IL, USA), distributed by the software company International Business Machines Corporation (IBM, Armonk, NY, USA), for further descriptive and comparative statistical analysis. Frequencies were reported in both absolute numbers and percentages, rounded to one decimal place. Categorical variables were compared using the chi-square test, and means were compared with Student’s *t*-test. To further investigate the possible relationships between the variables in the violence group, variables were examined by means of a correlation analysis (phi coefficient (ϕ) and contingency correlation coefficient, respectively). A two-sided *p* value of <0.05 was assumed to be statistically significant. Metrically scaled data are expressed as the arithmetic mean ± standard deviation (SD).

### 2.2. Graphical Presentation of the Results

Figure 1 was created using the commercial software Microsoft Excel. The Trauma Registry (TR) of the German Society for Trauma Surgery (DGU^®^) publishes a detailed annual report on serious traumatological injuries in Germany, Switzerland and Austria (DGU^®^; Annual Report 2021 TR-DGU, http://www.traumaregister.de; accessed on 1 March 2023). Among other things, the injury frequency and the injury severity of the recorded trauma patients are shown in a similar figure with similar color coding. Based on this figure, Figure 2 was designed. The figure was made using the commercial software Adobe Illustrator (San José, CA, USA). Copyright is held by the corresponding author.

## 3. Results

A total of 645 patients were evaluated. Of these, 290 patients form the after-violent-confrontation group (the violence group, VG) and the remaining 355 patients form the comparison group (CG). The trauma ED treats over 9000 patients per year, therefore the 290 patients treated after violent confrontation represent approximately 3.5% of the total trauma collective.

### 3.1. Reason for Presentation

Leading reasons for presentation in the CG were distortion trauma (22.8%), falls (outdoor, 16.1%; domestic, 10.4%) and impact trauma (10.4%). The remaining reasons for presentation are shown in Table 1.

### 3.2. Demographics

In the VG, 82.8% (*n* = 240) of patients were male compared to 53.8% (*n* = 191; *p* < 0.000) in the CG. The mean age was 32 years (±10.4 years SD) in the VG and 35 years (±20.9 years SD; *p* = 0.107) in the CG (Table 2).

### 3.3. Presentation Time

In the VG, 6.9% (*n* = 20) of patients presented on a Monday compared to 13% (*n* = 46; *p* = 0.051) for the CG; 10.3% (*n* = 30) of VG patients presented on a Tuesday compared to 11.8% (*n* = 42; *p* = 0.635) for the CG; 9% (*n* = 26) of VG patients presented on a Wednesday versus 15.2% (*n* = 54; *p* = 0.063) for the CG; 7.6% (*n* = 22) of VG patients presented on a Thursday versus 11.8% (*n* = 42; *p* = 0.162) for the CG; and 12.4% (*n* = 36) of VG patients presented on a Friday compared to 17.2% (*n* = 61; *p* = 0.185) for the CG. With regard to the weekend days of Saturday and Sunday: 23.4% (*n* = 68) of VG patients presented on Saturdays compared to 17.5% (*n* = 62) of CG patients (*p* = 0.123); and 30.3% (*n* = 88) of VG patients presented on Sundays compared to 13.5% (*n* = 48) of CG patients, which is a significant difference (*p* < 0.000). The weekday presentation times for both groups are shown in Figure 1.

Overall, weekend presentations (defined as the period from 06:00 p.m. Friday to 06:00 a.m. Monday) occurred significantly more often in the VG compared to the CG (62.8%, *n* = 182 vs. 37.2%, *n* = 132; *p* < 0.000). In addition, the VG also showed significantly more frequent presentations at night (defined as the period between 10:00 p.m. and 06:00 a.m.) compared to the CG (75.9%, *n* = 220 vs. 11.3%, *n* = 40; *p* < 0.000) (Table 2). Compared to daytime and weekday differences in emergency department presentations, no significant differences could be found with regard to the seasons and also on special holidays such as New Year’s Day, Labor Day or on special major events that took place in Frankfurt am Main.

### 3.4. Pre-Clinical and Clinical Assessment

Compared to the CG, significantly more patients in the VG presented to the ED via the ambulance service [67.6% (*n* = 196) of VG vs. 23.1% (*n* = 82) of CG; *p* < 0.000]. Similarly, admission via the trauma room was significantly more frequent in the VG than in the CG [10.3% (*n* = 30) of VG vs. 1.7% (*n* = 6) of CG; *p* < 0.000]. Upon presentation to the ER, routinely drawn blood samples showed laboratory detection of a positive blood alcohol concentration in 50.3% (*n* = 146) of patients in the VG. In the VG, radiological imaging was performed for further diagnosis after physical examination in 74.5% of cases (*n* = 216), compared to 79.2% in the CG (*n* = 281; *p* = 0.254). Computed tomography (CT) scans were performed significantly more often in the VG [41.4% (*n* = 120) of VG vs. 9% (*n* = 32) of CG; *p* < 0.000]. The need for surgical wound care in the ED for patients in the VG was also significantly more frequent than in the CG [49% (*n* = 142) of VG vs. 14.7% (*n* = 52) of CG; *p* < 0.000]. Regarding the need for hospitalization of the patients, there was no significant difference between the two groups [12.4% (*n* = 36) of VG vs. 7.9% (*n* = 28) of CG; *p* = 0.112]. There was also no significant difference between the groups in the need for surgical treatment of the injury sustained [4.8% (*n* = 14) of VG vs. 5.4% (*n* = 19) of CG; *p* = 0.811] (Table 2).

### 3.5. Injury Pattern and Discharge Diagnosis

Patients in the VG sustained injuries to the head significantly more often than those in the CG [53.8% (*n* = 156) of VG vs. 7.9% (*n* = 28) of CG; *p* < 0.000]. Injuries to the trunk did not show significant differences. In contrast, both, upper and lower extremity injuries were significantly more frequent in the CG than in the VG [upper extremity: 22.8% (*n* = 66) of VG vs. 38% (*n* = 135) of CG; *p* = 0.001; lower extremity: 5.5% (*n* = 16) of VG vs. 38.3% (*n* = 136) of CG; *p* < 0.000]. Figure 2 shows comparatively the injured body regions of the two groups.

The discharge diagnosis “wound” was significantly more frequent in the VG than in the CG [46.9% (*n* = 136) of VG vs. 7.9% (*n* = 28) of CG; *p* < 0.000] as well as contusions (42.1%, *n* = 122 vs. 30.4%, *n* = 108; *p* = 0.012). Craniocerebral traumas were detected significantly more frequent in the VG (5.5%, *n* = 16) than in the CG (1.7%, *n* = 6; *p* = 0.019). On the other hand, fractures and distortions were rather visible in the CG group [fractures: 4.8% (*n* = 14) of VG vs. 18.9% (*n* = 67) of CG; *p* < 0.000; distortions: 0% (*n* = 0) of VG vs. 22% (*n* = 78) of CG; *p* < 0.000]. All data at discharge are shown in Table 3.

### 3.6. Correlation

In order to further investigate potential relationships between the variables assessed in the violence group, they were examined by means of correlation analysis. Here, male gender was found to be significantly correlated with positive alcohol detection (ϕ = 0.314, *p* < 0.001). In addition, positive alcohol detection was significantly correlated with presentations at night (ϕ = 0.246, *p* < 0.001) and with presentations at weekends (ϕ = 0.177, *p* = 0.003). Also, positive alcohol detection was correlated with injuries to the head (contingency coefficient = 0.302, *p* < 0.001) and with the need for surgical wound care (ϕ = 0.437; *p* < 0.001). Male gender was also correlated with injuries to the head (contingency coefficient = 0.305, *p* < 0.001). No significant correlations were found for other variable combinations.

## 4. Discussion

The data shown in this paper demonstrate that the care of patients after violent confrontation represents a visible proportion in the trauma ED. The injury pattern clearly differs in many aspects from the “normal” injured patients. We found out that patients in the VG predominantly suffered injuries in the area of the head. Concomitantly, more than 50% of patients in the VG were under the influence of alcohol. Due to the coincidence of head injury and alcohol intoxication, adequate assessment of neurological status by the examiner is often difficult, as both head trauma and acute alcohol intoxication may be accompanied by loss of consciousness, amnesia, nausea/vomiting or disorientation [5,6]. In clinical practice, the Glasgow Coma Scale (GCS) in particular has been used to assess the state of consciousness in patients with traumatic brain injury for decades [7]. Whether and to what extent acute alcohol intoxication has a relevant impact on the GCS is still controversial in the existing literature [8,9]. However, our own clinical experience reveals that assessment of the neurological condition of the intoxicated patient is often complicated by the fact that medical history questions are not answered adequately and the patients behave incompliantly during the physical examination. This may also explain the more frequent admission of patients in the VG compared to the CG via the trauma room in this study. In the case of neurological status that is not reliably assessable, along with frequently existing bleeding wounds in the area of the head (in the VG in almost 50% of the cases), patients may be presented via the trauma room for faster exclusion of intracranial pathologies, which in turn may lead to overtriage of this patient collective. Moreover, the CT scan is often of limited quality due to agitation and incompliance of the intoxicated patients. A paper by Weber et al. showed that motion artifacts occurred in 27% of the cranial CT scans performed in patients under the influence of alcohol [10]. In clinical practice, additional sedation of these patients is therefore necessary in order to perform adequate imaging. This in turn may also be associated with delayed diagnostics and prolonged use of trauma room staff, leading to their unavailability for other (real) emergencies.

In the VG, over 40% of patients received CT imaging, the majority of which was CT of the head. A study by Paul et al. in the USA showed that the total price for performing a CT of the head in academic hospitals averaged $1390.12 (±$686.13 SD), highlighting CT imaging as a relevant cost factor in the healthcare system [11].

Evidence of intracranial hemorrhage was found in six out of 290 patients (2.1% of cases) in this study, as compared to two patients in the controls. Consistent with this, a study by Godbout et al. reported the detection of intracranial hemorrhage in 1.9% of all alcohol-intoxicated patients examined in the ED [12]. By contrast, a study by Easter et al. showed that 8% of alcohol-intoxicated patients with head injuries sustained clinically relevant intracranial injuries [13]. The higher number of relevant intracranial injuries compared with the present work may be explained by the mechanism of the injury sustained. Thus, in the study by Easter et al., falls and traffic accidents (in terms of high-energy trauma) contributed causally to the intracranial injuries in addition to violent confrontation. Recently, our research group showed that stair falls were associated with intracranial hemorrhage and alcohol intoxication [14]. Easter et al. also highlighted that accepted clinical decision support tools as indicators for considering cerebral CT imaging, such as the Canadian CT Head Rule [15] or National Emergency X-Radiography Utilization Study criteria [16], do not have sufficient sensitivity for predicting clinically relevant intracranial injury in a cohort of alcohol-intoxicated patients. Therefore, the indication threshold for performing CT in the presence of concomitant head injury and alcohol intoxication should be set rather low in order not to overlook intracranial trauma.

Almost half of the patients suffered head injuries with accompanying head lacerations requiring surgical wound care. Previously, our research group was able to show that surgical wound care is financially deficient under the current German remuneration system [17]. In particular, patients under the influence of alcohol require more time due to the difficulty of taking their medical history, undergoing physical examination and further diagnostics, and this is accompanied by a prolonged commitment of resources [18]. Furthermore, as shown in the present study, this group of patients mostly presents to the ED at night and on weekends. In addition, patients in the VG were significantly more likely to be admitted by ambulance service or trauma room, which additionally represents a cost burden for the healthcare system. A study from our working group showed that for those patients admitted via the trauma room and suffering only minor craniocerebral trauma, the cost coverage was not achieved in a single case, despite the fact that relevant items such as material and personnel costs were not even included in the calculation [19].

Men were significantly overrepresented in the present study, accounting for over 80% of cases. It should be mentioned, however, that women who presented to the ED in the context of domestic violence were excluded from this study, which somewhat limits the power of the data. Numerous studies in the past have documented a generally higher involvement of men in violence, although gender differences in propensity to violence appear to have neurobiological as well as sociocultural causes and remain the subject of current research [20]. In addition to male gender, presentations after violent confrontation were also frequently associated with alcohol use. Men misuse alcohol more often than women and are almost twice as likely to engage in binge drinking [21]. In the context of excessive alcohol consumption, both aggression and the risk of physically assaulting another person increase [22]. Consequently, this also increases the risk for staff working in the ED to be victims of verbal abuse and/or physical assault [23,24]. In the healthcare sector, the ED is thereby most often affected by violent behavior from patients [25]. This can lead to decreased job satisfaction and work effectiveness, as well as increased sick leave among ED staff, which in turn can negatively impact patient quality of care [26]. Although the evidence from studies that examined the effectiveness of interventions to promote a safe work environment in the ED is limited [27], regular violence prevention training or education for the staff [28,29] but also the permanent deployment of security personnel [30] seem to promote a more secure workplace environment in the ED.

## 5. Conclusions

The present study demonstrates that presentations in the ED after a violent confrontation account for a relevant proportion of treatment by the ED personnel. The examined patient collective showed primarily injuries of the head, with severe intracranial injuries being the exception. Nevertheless, due to the high coincidence of head injuries and alcohol intoxication, there should be neither an underestimation of injury severity nor a delay or omission of important diagnostic measures. Since these patients are particularly often admitted via the ambulance service or trauma room and, overall, more diagnostic CT imaging was performed, these cases represent a relevant cost factor for the healthcare system. It may be necessary in the future to discuss an adjustment of the remuneration or additional compensation for this patient group in order to ensure sufficient quality of care. Due to a steadily increasing number of patients, as well as existing staff and resource shortages and a resulting high workload in the ED, it must also be ensured that externally aggressive behavior towards ED staff is reduced as an additional stressor by, for example, a constant presence of security staff and offers of violence prevention training programs.

### Limitations

Although the present study, due to its retrospective nature, may not capture all cases in the selected observation period, the gathered data demonstrate (clinical) relevance for the treating ED staff. The estimated number of unreported cases for presentations after violent confrontation is probably even higher, as many patients, possibly out of shame, do not mention that they suffered the injuries resulting in their presentation to the ED in the context of a violent act. Compared to other locations, Frankfurt University Hospital may be particularly exposed to presentations following physical altercations due to its close local relationship to Frankfurt′s main train station, which enjoys a special image across Germany because of its open drug trafficking and drug use. From the mid-1970s onward, there was a rapid increase in the public use of hard drugs such as heroin and crack in Frankfurt′s urban area. In the 1980s and 1990s, the core of the open drug scene then established itself in the immediate vicinity of the main train station, where prostitution is also widespread [31]. A direct comparison with other metropolitan hospitals would, then, also be interesting here.

## Figures and Tables

**Figure 1 ijerph-20-04552-f001:**
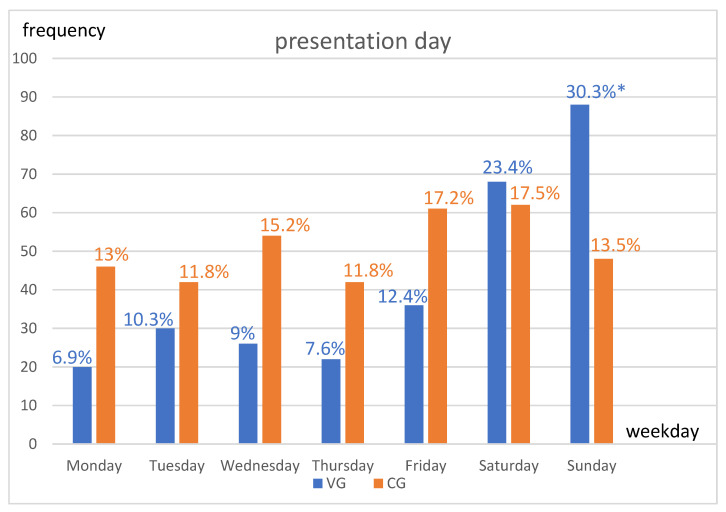
Shown are frequencies (including percentages) of presentations by day of week in direct comparison of groups [blue = violence group (VG); orange = comparison group (CG)]. The chi-square test was used for the statistical analysis. * *p* < 0.05.

**Figure 2 ijerph-20-04552-f002:**
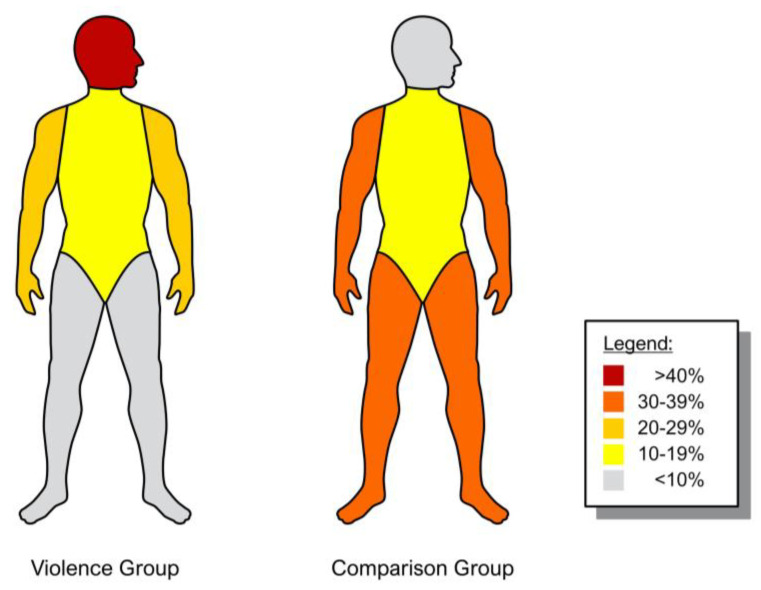
Shown are the frequencies (in percent) of injuries to the respective body regions affected in a direct comparison of the groups (left = violence group; right = comparison group). The trunk region includes the entire spine, thorax, abdomen and pelvis. Copyright is held by the corresponding author.

**Table 1 ijerph-20-04552-t001:** Reason for presentation of the comparison group.

Reason for Presentation	Frequency *n* (Total *n* = 355)
wound	*n* = 25 (7.0%)
impact trauma	*n* = 37 (10.4%)
distortion trauma	*n* = 81 (22.8%)
bruising trauma	*n* = 25 (7.0%)
domestic fall	*n* = 37 (10.4%)
outdoor fall	*n* = 57 (16.1%)
bicycle fall	*n* = 15 (4.2%)
traffic accident	*n* = 26 (7.3%)
redness/swelling (limb)	*n* = 12 (3.4%)
pain without trauma	*n* = 20 (5.6%)
other	*n* = 20 (5.6%)

Shown are the different reasons that led to a presentation of the patients of the comparison group in the emergency department. Among others, “other” reasons for presentation include lifting trauma, burns/scalds or needlestick injuries.

**Table 2 ijerph-20-04552-t002:** Demographics, preclinical and clinical characteristics.

	Violence Group (*n* = 290)	Comparison Group (*n* = 355)	*p*-Value
age (years)	32.1 (±SD 10.4)	35 (±SD 20.9)	0.107
sex (male)	*n* = 240 (82.8%)	*n* = 191 (53.8%)	<0.000 *
ambulance service	*n* = 196 (67.6%)	*n* = 82 (23.1%)	<0.000 *
trauma room	*n* = 30 (10.3%)	*n* = 6 (1.7%)	<0.000 *
weekend	*n* = 182 (62.8%)	*n* = 132 (37.2%)	<0.000 *
presentation at night	*n* = 220 (75.9%)	*n* = 40 (11.3%)	<0.000 *
imaging performed	*n* = 216 (74.5%)	*n* = 281 (79.2%)	0.254
CT performed	*n* = 120 (41.4%)	*n* = 32 (9%)	<0.000 *
wound care	*n* = 142 (49%)	*n* = 52 (14.7%)	<0.000 *
inpatient admission	*n* = 36 (12.4%)	*n* = 28 (7.9%)	0.112
operation	*n* = 14 (4.8%)	*n* = 19 (5.4%)	0.811
alcohol	*n* = 146 (50.3%)	no data	

Shown are demographic as well as preclinical and clinical characteristics in a direct comparison of the groups. Abbreviations: CT: computed tomography, SD: standard deviation. Since blood sampling is not standard practice in every patient, blood alcohol concentrations are not available for the comparison group. Student’s *t*-test was used to compare the mean age values. For the remaining categorical variables, the chi-square test was used. * *p* < 0.05.

**Table 3 ijerph-20-04552-t003:** Discharge diagnosis.

	Violence Group (*n* = 290)	Comparison Group (*n* = 355)	*p*-Value
wound	*n* = 136 (46.9%)	*n* = 28 (7.9%)	<0.000 *
contusion	*n* = 122 (42.1%)	*n* = 108 (30.4%)	0.012 *
fracture	*n* = 14 (4.8%)	*n* = 67 (18.9%)	<0.000 *
dislocation	*n* = 2 (0.7%)	*n* = 8 (2.3%)	0.233
TBI	*n* = 16 (5.5%)	*n* = 6 (1.7%)	0.019 *
distortion	*n* = 0 (0%))	*n* = 78 (22%)	<0.000 *
other		*n* = 60 (16.9%)	

The frequencies of discharge diagnoses are shown in a direct comparison of the groups. Among others, “other” diagnoses include polytrauma, burns/scalds, and erysipelas/phlegmons/abscesses. Abbreviations: TBI: traumatic brain injury. The chi-square test was used for the statistical analysis. * *p* < 0.05.

## Data Availability

Data is contained within the article.

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
