# Peer review of "Saturday Night Fever: Interpersonal Violence as a Reason for Presentation in a University Emergency Department"

_ijerph, 2023, doi:10.3390/ijerph20054552_

Round 1
Reviewer 1 Report
The paper by Hörauf et al. represents a retrospective study of 645 patients (290 patients in violent confrontation and 355 in the control group) presenting to the Emergency Department of the University Hospital Frankfurt am Main between 1 January and 31 December 2019. As such, it is a welcomed addition to the literature within the field as it initiates important conversations about interpersonal violence incidence in EDs and safety of healthcare providers. With that, this study also points towards inadequate violence prevention training for healthcare providers wherein they might not always be equipped to process such cases in a safe manner.
The Introduction section is well written and provides the reader with an adequate overall review of the research within the field. The authors offer a brief general overview of some studies researching interpersonal violence in the domestic setting and point out the lack of representative data on interpersonal violence in a broader context. This is sufficient motivation for conducting this study.
The Materials and Methods section is also very well written, including well-defined inclusion and exclusion criteria. What is especially commendable here is the inclusion of a comparison group. The statistical analysis section is well presented and adequate for this type of data.
The Results section contains useful information and general overview demographics, presentation time and pre-clinical and clinical assessment of patients. Still, the authors only chose to include “reasons for presentation” of patients in the comparison group. Since this group was chosen so as to allow for comparison with patients admitted due to violent confrontations, it would be useful to also include a similar table for the VG. Alternatively, the authors could just add a column including frequency of these reported within the VG. Moreover, even though the subsection 3.5 includes a very interesting and helpful figure (Fig. 2) depicting the frequencies of injuries to the respective body regions for VG and CG, it is unclear as to what authors meant by “The figure shown has been modified based on the illustration in the annual report of the TraumaRegister (TR) of the German Society for Trauma Surgery (DGU®; Annual Report 2021 TR-DGU, http://www.traumaregister.de).” As such, I would suggest that the authors include a sentence or two discussing how this figure was produced within a relevant paragraph.
The Discussion section is extremely elaborate and cohesive wherein the authors put forth some very interesting observations. What is particularly commendable is the fact that the authors clearly recognize one major limitation of this study and that is the exclusion of women who presented to the ED in the context of domestic violence. This was a reasonable exclusion criterion since the authors wished to assess other incidences of interpersonal violence on which less is written within the literature. Moreover, of specific interest are the lines 241 – 249, wherein the authors discuss the risks to ED workers that come with treating patients following violent incidences. This is also further reinstated within the Conclusion section (lines 261 – 264) wherein the authors suggest inclusion of additional violence prevention training programs for healthcare workers. Finally, even though the Limitations of the study are well recognized, I would suggest that the authors find a reference, or further elaborate on their statement, pertaining to lines 273 and 274 and drug trafficking/use. Even though I acknowledge and appreciate that the authors have a good subjective overview of this, as they are present and aware of their surroundings, a reference here would be welcomed. It would provide a reader who is unfamiliar with the social context, additional reading to support this statement.
Ultimately, even though the study has been conducted in an adequate manner, it is lacking in some respects. Namely, since the authors have access to data pertaining to patients who underwent violent altercations throughout the period of a full year, it would be interesting to see whether these outbursts and, as such, patients’ presentation to ED, changes throughout this period of time. As such, I suggest the authors re-analyse the data and comment on whether certain dates/seasons correlate with an increase in admission of patients who took part in violent outbursts. For example, violence appears to be more common on holidays and vacation days, such as New Year and Labor Day (as most people are off from work and consume excessive amounts of alcohol), but also during the hot months (see https://doi.org/10.1016/S2542-5196(21)00210-2). With that, the authors would take a greater advantage of the plethora of data they have access to and contribute, to a greater extent, to the current state of the art. Therefore, I suggest the publication of this MS in IJERPH following major revisions.
Author Response
Dear Reviewer,
Thank you for reviewing our manuscript “Saturday night fever: Interpersonal violence as a reason for presentation in a university emergency department”, and for all the valuable comments. We agree with the comments and we have addressed them below. Changes have been made to reflect this, and the revised manuscript with highlighted changes has been attached.
Point 1: Still, the authors only chose to include “reasons for presentation” of patients in the comparison group. Since this group was chosen so as to allow for comparison with patients admitted due to violent confrontations, it would be useful to also include a similar table for the VG. Alternatively, the authors could just add a column including frequency of these reported within the VG.
Response 1: The group of patients presenting to our emergency department after a violent confrontation should be presented and investigated as a separate "presentation entity". In order to provide the readers of this paper with a more detailed description of a "typical" trauma patient population presenting to the emergency department, the reasons for presentation have been differentiated in more detail here (distortion trauma, for example, during sports or normal walking, bicycle falls, traffic accidents, etc.). However, since the categories (e.g. wound, impact trauma/contusion, distortion) between presentation reason and discharge diagnosis (see table 3) overlap to a large extent, the frequency between the groups can be indirectly differentiated. Thus, in the comparison group, the (primary) reason for presentation with a wound (for example, cuts or other (chronic) wounds) was significantly less frequent than in the violence group. Note the discrepancy in the comparison group between the presentation reason "wound" (table 1, n=25) and the discharge diagnosis "wound" (table 3, n=28). This can be explained by the fact that patients with a serious bicycle accident were assigned to the section "bicycle fall" for the presentation reason, but, for example, sustained a severe wound without a fracture or something similar. "Wound" was then coded as the primary discharge diagnosis. Therefore, the frequency of the category "wound" in the comparison group occurs more often in the "discharge diagnosis" section than in the "reason for presentation" section.
Point 2: Moreover, even though the subsection 3.5 includes a very interesting and helpful figure (Fig. 2) depicting the frequencies of injuries to the respective body regions for VG and CG, it is unclear as to what authors meant by “The figure shown has been modified based on the illustration in the annual report of the TraumaRegister (TR) of the German Society for Trauma Surgery (DGU®; Annual Report 2021 TR-DGU, http://www.traumaregister.de).” As such, I would suggest that the authors include a sentence or two discussing how this figure was produced within a relevant paragraph.
Response 2: The Trauma Registry of the German Society for Trauma Surgery publishes a detailed annual report on serious injuries in Germany, Switzerland and Austria. Among other things, the injury frequency and the injury severity of the recorded trauma patients are shown in a similar figure with similar color coding. Based on this figure, the present figure (Fig. 2) was designed. The figure was made using the commercial software Adobe Illustrator (San José, California, United States). Copyright is held by the corresponding author. This has been added to the Material and Methods section in lines 91-116.
Point 3: Finally, even though the Limitations of the study are well recognized, I would suggest that the authors find a reference, or further elaborate on their statement, pertaining to lines 273 and 274 and drug trafficking/use. Even though I acknowledge and appreciate that the authors have a good subjective overview of this, as they are present and aware of their surroundings, a reference here would be welcomed. It would provide a reader who is unfamiliar with the social context, additional reading to support this statement.
Response 3: From the mid-1970s onward, there was a rapid increase in the public use of hard drugs such as heroin and crack in Frankfurt's urban area. In the 1980s and 1990s, the core of the open drug scene then established itself in the immediate vicinity of the main train station, where prostitution is also widespread. To help readers understand the social context, this has been explained in more detail in lines 334-338.
Point 4: Ultimately, even though the study has been conducted in an adequate manner, it is lacking in some respects. Namely, since the authors have access to data pertaining to patients who underwent violent altercations throughout the period of a full year, it would be interesting to see whether these outbursts and, as such, patients’ presentation to ED, changes throughout this period of time. As such, I suggest the authors re-analyse the data and comment on whether certain dates/seasons correlate with an increase in admission of patients who took part in violent outbursts. For example, violence appears to be more common on holidays and vacation days, such as New Year and Labor Day (as most people are off from work and consume excessive amounts of alcohol), but also during the hot months.
Response 4: Thank you very much for the good advice. In the course of the data analysis, we examined whether, in addition to a daytime and weekday clustering of patients' presentations after a violent confrontation, there was also a seasonal clustering. However, this could not be confirmed. Thus, a total of 139 patients (47.9%) presented in summer (defined as the months of May to October) and 151 patients (52.1%) presented in winter (defined as the months of November to April). Also, after a re-examination of the data, no clustering of violent presentations could be found on special holidays such as New Year or Labor Day, respectively on major events such as the annual World Club dome music festival in Frankfurt. However, in a not yet published study of our research group, we were able to show that, overall, admissions to the emergency department of alcoholized patients are more frequent on such big event days in Frankfurt. In these cases, minor traumas such as twisted ankles, falls, etc. often predominate. The statement that there were no seasonal differences as well as no differences in presentation after violent confrontation on common vacations days and holidays such as New Year and Labor day was added in lines 162-166.

Reviewer 2 Report
1. From section 2 (Materials and Methods), I am not clear about the statistical sampling method the authors have used for selecting participants, please explain with clarity. I am also confused about how authors are confident about their sample size. Have you used any validation tests based on your selected sample size?
2. I am not clear about the significance of this study compared with existing studies. Please clarify it.
3. The author has included some statistical tests, but they have not included any statistical model for explaining all variable's effects. If it is possible include a statistical model in your analysis.
4. Need to be a clearer definition of violence. Based on which criteria you are defining the patients’ activities as violent?
Author Response
Dear Reviewer,
Thank you for reviewing our manuscript “Saturday night fever: Interpersonal violence as a reason for presentation in a university emergency department”, and for all the valuable comments. We agree with the comments and we have addressed them below. Changes have been made to reflect this, and the revised manuscript with highlighted changes has been attached.
Point 1: From section 2 (Materials and Methods), I am not clear about the statistical sampling method the authors have used for selecting participants, please explain with clarity. I am also confused about how authors are confident about their sample size. Have you I used any validation tests based on your selected sample size?
Response 1: Thank you for these important comments. The aim of this study was to characterize the patient population presenting to our emergency department after a violent confrontation and to compare it with a "typical" traumatological patient cohort (e.g. distortion trauma, ground level falls, bycicle accidents, etc.) and to identify possible differences. Therefore, a total of more than 9000 admission records were retrospectively searched and analyzed individually, which are generated standardly by the trauma surgeon on duty in the emergency department for each admitted patient. In each admission record, the reason for presentation is documented. If the admission record documented that the presentation was due to a violent confrontation (excluding women after domestic violence), the respective patient was included in the violence group. More specifically, patients were included in the violence group if they sustained visible injuries (e.g., wounds, bruises, hematomas) or demonstrable injuries (e.g., tenderness of the chest) on clinical examination in the course of a reported (self-reported or third-party) physical confrontation. Thus, a total of 290 patients were included in the violence group. If a violent confrontation was recorded on a particular day, all other admission records generated for that day were also captured. These then served as a data pool for the comparison group. The variables were then tested graphically for normal distribution using a Q-Q plot. To increase the robustness of the results, several random samples were then drawn from this data set using statistical software SPSS and the results subsequently were validated with the other parts. From the data pool, a random sample of 355 patients was defined as the comparison group using SPSS. A more detailed explanation regarding the statistical sampling method has been added in the "material and methods" section in lines 56-61 and 65-75.
Point 2: I am not clear about the significance of this study compared with existing studies. Please clarify it.
Response 2: Existing literature has specifically examined domestic violence against women respectively intimate partner violence. Recently, there has also been specific focus on the impact of violence on staff working in the emergency department. However, the exact patterns of injury sustained in the course of a violent confrontation, in particular, are much less frequently documented. A direct comparison to a typical traumatological patient population has also been lacking in the literature. In particular, the present study revealed the exact pattern of injuries and which diagnostic and therapeutic measures were required. Although no detailed cost-revenue calculation was performed in this study, it can be assumed that the treatment of patients after an assault is monetarily loss-making due to the significantly more frequent use of CT diagnostics and wound care, and that this should possibly be adjusted in the remuneration in the future.
Point 3: The author has included some statistical tests, but they have not included any statistical model for explaining all variable's effects. If it is possible include a statistical model in your analysis.
Response 3: As the present work is a restrospective study, an exact statistical model for the calculation of the number of cases cannot be applied. Nevertheless, for a better comparison of the statistical effects, we have chosen a similarly designed control group of patients (see above) in order to be able to show in which areas certain differences exist. To further examine the possible associations between the variables in the violence group, the present dataset was now additionally examined by means of correlation analysis. The results obtained are added to the manuscript in lines 218-228.
Point 4: Need to be a clearer definition of violence. Based on which criteria you are defining the patients’ activities as violent?
Response 4: Patients were included in the violence group if they suffered visible injuries (e.g., wounds, bruises, hematomas) or injuries that were detectable on clinical examination (e.g., tenderness of the thorax) in the course of a reported (self-reported or third-party) physical confrontation. A more precise definition of violent confrontation, which led to inclusion in the violence group, was added in lines 56-61.

Reviewer 3 Report
It was an interesting study, but it had many limitations
Although the study is retrospective and has a control group, proper discussion has not been done, and it is more similar to a descriptive study. If correlation or regression were done between factors such as alcohol consumption, drug use, etc., the results would be more interesting.
How did you calculate the sample size?
Please mention the statistical tests used in each table below the table
Additional Comments
1. What is the main question addressed by the research?
the interpersonal violence incidents at an urban university hospital that led to a presentation/admission to the ED
2. Do you consider the topic original or relevant in the field?
Somehow
Does it address a specific gap in the field?
Somehow
3. What does it add to the subject area compared with other published material?
nothing
4. What specific improvements should the authors consider regarding the methodology? What further controls should be considered?
The correlation between data, for example between the injury and alcohol and drug use, gender and injury, time of accident and injury location and ...
5. Are the conclusions consistent with the evidence and arguments presented and do they address the main question posed?
yes
6. Are the references appropriate?
yes
7. Please include any additional comments on the tables and figures.
nothing
Author Response
Dear Reviewer,
Thank you for reviewing our manuscript “Saturday night fever: Interpersonal violence as a reason for presentation in a university emergency department”, and for all the valuable comments. We agree with the comments and we have addressed them below. Changes have been made to reflect this, and the revised manuscript with highlighted changes has been attached.
Point 1: Although the study is retrospective and has a control group, proper discussion has not been done, and it is more similar to a descriptive study. If correlation or regression were done between factors such as alcohol consumption, drug use, etc., the results would be more interesting.
Response 1: Thank you for this valuable comment. We fully agree with the reviewer that a closer examination is warranted regarding a possible dependence of the variables under study. We have therefore examined the present dataset by means of correlation analysis. The results obtained are added to the manuscript in lines 218-228.
Point 2: How did you calculate the sample size?
Response 2: Thank you for this important question. The aim of this study was to characterize the patient population presenting to our emergency department after a violent confrontation and to compare it with a "typical" traumatological patient cohort (e.g. distortion trauma, ground level falls, bycicle accidents, etc.) and to identify possible differences. Therefore, a total of more than 9000 admission records were retrospectively searched and analyzed individually, which are generated standardly by the trauma surgeon on duty in the emergency department for each admitted patient. In each admission record, the reason for presentation is documented. If the admission record documented that the presentation was due to a violent confrontation (excluding women after domestic violence), the respective patient was included in the violence group. More specifically, patients were included in the violence group if they sustained visible injuries (e.g., wounds, bruises, hematomas) or demonstrable injuries (e.g., tenderness of the chest) on clinical examination in the course of a reported (self-reported or third-party) physical confrontation. Thus, a total of 290 patients were included in the violence group. If a violent confrontation was recorded on a particular day, all other admission records generated for that day were also captured. These then served as a data pool for the comparison group. The variables were then tested graphically for normal distribution using a Q-Q plot. To increase the robustness of the results, several random samples were then drawn from this data set using statistical software SPSS and the results subsequently were validated with the other parts. From the data pool, a random sample of 355 patients was defined as the comparison group using SPSS. A more detailed explanation regarding the statistical sampling method has been added in the "material and methods" section in lines 56-61 and 65-75.
Point 3: Please mention the statistical tests used in each table below the table.
Response 3: The statistical tests used in each table have been added below the table.
Point 4: Additional comment number 4: The correlation between data, for example between the injury and alcohol and drug use, gender and injury, time of accident and injury location and ...
Response 4: See point/response 1.

Round 2
Reviewer 1 Report
The authors have meticulously addressed of my concerns through inclusion of additional details on the creation of Fig. 2, references for the relevant social context and have reflected on the lack of seasonal differences in patients' presentation to the ED. As such, they have greatly improved the quality of the manuscript and I suggest its publication in IJERPH.
Reviewer 2 Report
Thank you all authors for updating your article and clarifying queries according to my suggestions.
Reviewer 3 Report
Thanks for your answers.